# The Tbx6 Transcription Factor Dorsocross Mediates Dpp Signaling to Regulate *Drosophila* Thorax Closure

**DOI:** 10.3390/ijms23094543

**Published:** 2022-04-20

**Authors:** Juan Lu, Yingjie Wang, Xiao Wang, Dan Wang, Gert O. Pflugfelder, Jie Shen

**Affiliations:** 1MOA Key Laboratory of Surveillance and Management for Plant Quarantine Pests, Department of Plant Biosecurity, College of Plant Protection, China Agricultural University, Beijing 100193, China; juanlu2022@cau.edu.cn (J.L.); 12031186@mail.sustech.edu.cn (Y.W.); wangxiaonxy@126.com (X.W.); 2Institute of Developmental Biology and Neurobiology, Johannes Gutenberg-University, 55128 Mainz, Germany; pflugfel@uni-mainz.de

**Keywords:** thorax closure, decapentaplegic, *Dorsocross*, *Drosophila*, notum

## Abstract

Movement and fusion of separate cell populations are critical for several developmental processes, such as neural tube closure in vertebrates or embryonic dorsal closure and pupal thorax closure in *Drosophila*. Fusion failure results in an opening or groove on the body surface. *Drosophila* pupal thorax closure is an established model to investigate the mechanism of tissue closure. Here, we report the identification of T-box transcription factor genes *Dorsocross* (*Doc*) as Decapentaplegic (Dpp) targets in the leading edge cells of the notum in the late third instar larval and early pupal stages. Reduction of *Doc* in the notum region results in a thorax closure defect, similar to that in *dpp* loss-of-function flies. Nine genes are identified as potential downstream targets of Doc in regulating thorax closure by molecular and genetic screens. Our results reveal a novel function of Doc in *Drosophila* development. The candidate target genes provide new clues for unravelling the mechanism of collective cell movement.

## 1. Introduction

Closure of two separate epithelia is a common phenomenon in development and tissue repair of metazoans. It allows the formation of structures such as the vertebrate neural tube or embryo and thorax of *Drosophila*. Till now, over 200 genes have been reported to be involved in neural tube closure [1,2]. Movement and fusion of separate cell populations are critical processes for correct morphogenesis. Fusion failure results in an opening or groove on the body surface [3]. For example in humans, neural tube defect (NTD) is a common birth defect due to the failure of spinal column closure. NTD is characterized by a small or missing brain, or the exposure of the spinal cord [4]. *Drosophila* dorsal closure in the embryonic stage [5,6] and thorax closure in the pupal stage [7,8] are convenient models for investigating the mechanisms of tissue fusion processes. 

The adult thorax of *Drosophila* is derived from the notum parts of two larval wing imaginal discs [9]. Imaginal discs are invaginations of the larval epidermis made up of two opposing epithelia, a columnar main epithelium and a squamous peripodial epithelium. The differentiation into the two types of epithelium arises during larval development and requires continuous signaling input [10,11,12,13]. The two wing imaginal discs develop in spatial separation [14]. At the end of larval development, the discs evert [15,16] and the two heminota move toward the midline and fuse to form a complete thorax in the early pupal stage [8,17,18]. Cells at the junction between main and peripodial epithelium are of prime importance for the fusion process [17].

Decapentaplegic (Dpp), which is a member of the transforming growth factor-β superfamily, is essential for thorax closure. Disrupted Dpp signaling either at the level of its receptors Thickveins (Tkv) and Punt [8] or at the level of the signal transducer Medea [19] results in a thorax closure defect. Lack of Dpp signaling causes a block of filopodia extension and a collapse of the leading edge [7]. In the thorax, *pannier* (*pnr*) [20,21,22] and *eyegone* (*eyg*) [23] have been demonstrated to be regulated by Dpp in a concentration dependent manner. *pnr*, encodes a zinc-finger transcription factor of the GATA family, is positively regulated by Dpp signaling to specify the medial fate of notum. This fate specification is regulated either by activating another important morphogen Wingless (Wg) in the most lateral region or by repressing Wg in combination with transcriptional factor U-shaped in the medial region of notum [24]. Ectopic expression of *pnr* induces extra bristles which is a character of the thorax media patterning [21]. *eyg*, which encodes a homeodomain Pax protein, is expressed in the anterior region of notum by negative regulation of Dpp and positive regulation of Pnr [23]. The specific *eyg* expression in the anterior notum together with *drumstick* (*drm*) and *bowl* is required for the notum growth and patterning. Therefore, downstream factors mediating the role of Dpp signaling in thorax closure have not yet been identified.

Embryonic dorsal closure, another model for the standard epithelium fusion, occurs after germ band retraction with the dorsally directed migration of the lateral epidermis over the amnioserosa from both sides of the embryo [25]. Dpp signaling also regulates the dorsal closure by mediating the epidermis adhesion and the contraction of amnisoerosa [26]. The *Dorsocross* (*Doc*) genes, encode three functionally largely redundant T-box transcription factors Doc1, Doc2, and Doc3 of the Tbx6 subfamily [27,28,29,30,31], are expressed both in the embryonic dorso-lateral ectoderm and in the amnioserosa under the control of Dpp [30,32,33,34]. The three *Doc* genes are collectively referred to “*Doc* genes” based on the similarities sequence, expression patterns and genetic analysis. Loss of Doc perturbs amnioserosa function during germ band extension and retraction [30] such that a direct role of Doc in embryonic dorsal closure could not be determined. Expression and function of Doc extends beyond amnioserosa and the embryonic ectoderm to e.g., visceral and cardiac mesoderm development [32]. Right now, Doc2 antibody staining was used as a proxy to identify Doc protein expression pattern.

In the wing imaginal disc, there are four Doc expressing domains: two large domains located in the dorsal and ventral regions of the prospective wing blade and two smaller domains in the prospective dorsal hinge and posterior notum regions [30]. The two large wing pouch Doc expression domains are critical for the formation of a fold between hinge and blade regions of the larval disc and for wing disc bending [35]. However, the role of the smaller Doc expression domains is still unknown. Thoracic closure provides a setting in which the involvement of Doc in tissue fusion processes can be tested.

In this study, we investigate the role of *Doc* expression in the posterior notum. We show that this expression domain corresponds to the future leading edge of the fusion discs. *Doc* expression required Dpp and Pnr activity. When Doc was reduced in the leading edge cells of the notum, a thorax closure defect was induced similar to the phenotype caused by down-regulated Dpp signaling. Genetic analysis identified nine genes as downstream targets of Doc in regulating thorax closure. This novel function and identified candidates provide clues for future collective cell movement study. 

## 2. Results

### 2.1. Doc Is Expressed in the Leading Edge Cells during Thorax Closure

We and others have shown that, in late L3 wing discs, *Doc* expression includes a stripe of cells in the posterior medial notum [30,35]. We wondered whether this part of the *Doc* expression pattern was in the future leading edge cells. Thus, Doc antibody staining was performed from the mid 3rd larval instar to early pupal stages. There was no *Doc* expression in the posterior notum in the mid L3 wing disc (Figure 1A). *Doc* expression in the posterior notum was first detectable in late L3 wing discs (Figure 1B). In a cross-sectional view, it was obvious that *Doc* expression was restricted to the main epithelium and was excluded from the peripodial epithelium (Figure 1B’,B’’). At this stage, *Doc* and *puc-lacZ*[A251] were expressed along the posterior lateral region in a non-overlapping pattern (Figure 1C). During early pupal development (6 h After Puparium Formation, APF), *Doc* and *puc-lacZ*[A251] were co-expressed in the leading edge cells of the fusing heminota (Figure 1D–D’’). The enhancer trap line *puc-lacZ*[A251], which in the embryo faithfully reflects JNK pathway activity during dorsal closure [36,37], has a more restricted expression pattern than *puc-lacZ*[E69] in the wing disc [16,17,38], so that actual JNK pathway activity and *Doc* expression may overlap already in the L3 wing disc. Our results show how *Doc* expression expands from the late L3 to the early pupal stage, to finally take up the entire leading edge domain. The co-localization of *Doc* and *puc-lacZ* suggests a role of *Doc* in thorax closure. 

Since Dpp signaling is required for correct thorax closure [8,19], we tested for Dpp expression pattern and signaling in the leading edge cells during thorax closure. In tissue fusion, three steps have been discerned: a physiological and structural change of the leading edge cells, the joint movement of epithelia towards the midline, and fusion of the two separate epithelia [3,39,40]. We used *dpp-Gal4* driving *UAS-GFP* as a marker of *dpp* expression and *puc-lacZ*, to visualize the leading edge cells [8,17,38]. We found that *GFP* expression gradually expanded and co-localized with *puc-lacZ* in the leading edge cells from 5–6 h APF (Figure 1E,F). At 6–8 h APF, during which stage the heminota began to fuse, the *dpp > GFP* signal was still present and co-localized with *puc-lacZ* (Figure 1G). Similarly, a lacZ reporter of *dpp* was also observed in the leading edge cells (Figure 1H). The presence of Dpp signaling in leading edge cells was also demonstrated by PMad antibody staining (Figure 1I). From the above results, we conclude that Dpp signaling is present in the leading edge cells throughout the thorax closure process.

### 2.2. Doc Is Regulated by Dpp Signaling in the Notum Region of the Wing Disc

In the embryonic dorsolateral ectoderm, Doc acts downstream of Dpp signaling [34]. Given the co-expression of *Doc* and *dpp* in the wing disc notum [30,35] (Figure 2A), we asked whether *Doc* is a Dpp target during thorax closure. To block Dpp signaling, a dominant negative form of the receptor Tkv (*tkv^DN^*) [41] was expressed in the *ap-Gal4* domain. *Doc* expression was lost from the entire dorsal compartment including the presumptive leading edge cells (Figure 2B,B’). 

*pnr* is induced by Dpp in the embryonic dorsal ectoderm and is required for dorsal closure. However, after embryonic stage 10, *dpp* becomes Pnr-dependent in the dorsal ectoderm [42]. *pnr* is also induced by Dpp in the notum region of the wing disc and is required for dorso-ventral patterning [24,43]. Transheterozygous combinations of *pnr* alleles show thorax fusion defects [43]. By *pnr* knock-down, we showed that *pnr* was required for notal *Doc* expression (Figure 2C,C’). Another factor specifying notum development is *eyg*, which is induced by Pnr but repressed by Dpp [23]. Missexpression of *eyg* in the posterior notum region of the wing disc blocked *Doc* expression (Figure 2D,D’). Broad expression of *pnr* in the notum region of the wing disc did not suffice to extend the *Doc* expression domain, presumably due to maintained expression of antagonistic factors such as Eyg (Figure 2E). Co-expression of *pnr* was sufficient to rescue *Doc* repression caused by *tkv^DN^* expression (Figure 2F,G) suggesting that Pnr acts downstream of Dpp signaling in the induction of *Doc*.

However, ectopic expression of *dpp* (Appendix A) or knocking-down of *eyg* (Appendix A) had no effect on *Doc* expression in the wing disc. This implies other factors, in addition to Dpp signaling, regulate *Doc* expression in the wing disc notum region.

### 2.3. Doc Is Required for Correct Thorax Closure

The *Drosophila* adult notum exhibits a smooth surface with regularly arranged bristles (Figure 3A). Inhibition of Dpp signaling in the medial notum by *tkv^DN^* caused a thorax closure defect [8] (Figure 3B). A similar phenotype was obtained by knocking-down *pnr* (Figure 3C). Using the adult phenotype, we asked whether *Doc* plays role in thorax closure. Partial knock-down of *Doc* by *Doc1-RNAi* plus *Doc2-RNAi* (Appendix A) or by *Doc3-RNAi* (Figure 3D) resulted in a thorax closure defect, demonstrating a requirement for *Doc* in correct thorax closure. A quantification of defect frequency is shown in Appendix A. The phenotype penetrance between *Doc3-RNAi* expression and *Doc1* + *2-RNAi* co-expression was similar which indicates that the redundant role or the dosage effect may be involved in regulating thorax closure by *Doc* genes. Recovery of *Doc* by *pnr* expression was sufficient to rescue the thorax closure defect when Dpp signaling was suppressed (Figure 3E,F). The above results demonstrate that *Doc* acts downstream of Dpp signaling and is essential for thorax closure.

### 2.4. Functional Analysis of Potential Doc Target Genes

In a preliminary molecular screen, nine genes were selected for subsequent phenotype assays based on their differential expression levels in *Doc-RNAi* wing discs compared with wild type control (Appendix A). The 2 kb upstream of the nine genes’ promoter contained a sequence motif related to the *Doc* binding consensus TTCACACCT (Appendix A). In the following genetic analysis, seven genes (*arpc3B*, *GC12164*, *CG14456*, *CG16758*, *βtub97EF*, *attc*, *mec2*) which were downregulated in *Doc-RNAi* manipulation were investigated by crossing with *pnr-Gal4* for their effect on thorax development. Severe defects were observed when either of them was knocked-down by RNAi (Figure 4A–G). *pupal cuticle protein* (*pcp*) and *secreted decoy of InR* (*sdr*) were upregulated under *Doc-RNAi* treatment and were selected for gain-of-function assays. Overexpression of *pcp* or *sdr* under *pnr-Gal4* caused a mild thorax defect (Figure 4H,I). The thorax closure defect severity was evaluated by the gap between the two middle bristles. The distance of this gap was measured and treated as the thorax closure index which suggest significant difference between the gene manipulation and the wild type control (Figure 4J). Altogether, nine genes were identified as potential Doc target genes regulating thorax closure.

## 3. Discussion

The fusion of two separated epitheliums is occurred in nearly all the organisms which provides the possibility of forming a complete structure, such as the vertebrate neural tube, wound healing, complete *Drosophila* embryo and thorax. In this study, we present a genetic analysis of *Doc* function in *Drosophila* notum development to present a potential role of *Doc* in epithelium fusion. We show that *Doc* acts downstream of Dpp signaling in regulating thorax closure. Nine potential Doc target genes were identified in regulating thorax closure by an RNAi-based genetic screen with subsequent phenotypic analysis.

### 3.1. Dpp Signaling Promote Thorax Closure

*Drosophila* thorax closure requires numerous signal transduction pathways. Fos [7] and JNK [7,38], and Dpp signaling [8,19,44,45] are the best characterized pathways in this process. Dpp signaling was activated in the leading edge cells throughout the fusion process (Figure 1A–E). Inhibition of Dpp signaling resulted in a severe thorax closure defect (Figure 3A,B). *pnr* and *Doc*, acting as the Downstream of Dpp signaling, were essential for thorax closure (Figure 3C–I). 

In embryonic dorsal closure *dpp* is induced by JNK/Fos signaling [11,46,47,48,49,50,51,52]. Studies have shown that Ena [50], Cdc42 [53], Rac, and Rac-mediated JNK signaling [54] regulate the epithelial movement. Mutants defective in the JNK and Dpp pathways exhibit dorsal holes, and ectopic expression of an activated form of Tkv can rescue the defects induced by the loss of JNK signaling [51,55,56,57]. While the JNK and DPP pathways are linked, they are not functionally equivalent [57]. 

Dpp and JNK are also required for thorax closure. In contrast to dorsal closure, *dpp* is not controlled by JNK signaling in thorax closure [7,38]. Dpp is involved in the cell shape change of the leading edge and the spreading and adhesion of the two opposite cell sheets [8], while JNK appears predominantly involved in the leading edge cell shape changes [8]. However, Usui and Simpson [17] noted that in thorax closure, the JNK pathway was activated (*puc* expression) in two adjacent cell types differing in nuclear size which they termed S (talk) and I (imaginal cells). Based on observation, they attributed different function to S and I cells in the fusion process and assumed that I cells generate the force for disc migration. Doc-positive cells, judged by their location at the junction between main and peripodial epithelium (Figure 1B’) and their exclusion from the disc stalk (Figure 1C), can be considered as I cells. The fact that *Doc* knockdown caused a thorax fusion defect lends credit to the hypothesis of Usui and Simpson [17].

### 3.2. Doc Function in Other Epithelia

*Doc* function was first demonstrated in the embryonic dorso-lateral ectoderm and in the amnioserosa. *Doc* knock-down perturbs germband extension and retraction and amnioserosa development [30,32,33,34]. Loss of *Doc* perturbes amnioserosa function already during germ band extension and retraction [30]. These early effects precluded a direct analysis of *Doc* involvement in dorsal closure. A detailed analysis of the cell biological effects of *Doc* loss on germ band movement has not yet been performed. In the flour beetle *Tribolium castaneum*, the single *Tc-Doc* is not required for specification and maintenance of its amnion and serosa. This allowed investigation of whether and how *Tc-Doc* affects morphogenesis of the extraembryonic epithelia in this species. The first rearrangement of extraembyonic tissue (closure of the serosal window) is slowed down or blocked by *Tc-Doc-RNAi* [58]. *Doc* is also involved in the reorganisation of the *Drosophila* embryonic hindgut epithelium which initiates the formation of the four Malpighian tubles (MpT). The two pairs of MpTs develop from the hindgut at the mid/hindgut junction. *Doc* expression is restricted to where the anterior pair of MpTs buds from the hindgut. In *Doc* mutant embryos, development of the anterior pair of MpTs is suppressed while the posterior pair develops normally [59]. The precise role of *Doc* in MpT development is, however, not yet known [60]. We have previously analyzed the morphogenetic function of Doc in *Drosophila* wing imaginal disc development. *Doc* is expressed in the wing pouch (destined to develop into the wing blade) at the junction to the future wing hinge. These domains of the wing disc epithelium are separated by a deep fold. Doc promotes fold formation by inducing matrix metalloproteinase 2 expression and by causing a reorganization of the microtubule cytoskeleton [35]. Currently it cannot be judged whether Doc promotes restructuring of different epithelia via common targets.

### 3.3. Potential Doc Target Genes in Regulating Thorax Closure

Based on their Doc dependence and their adult notum phenotype, nine genes were identified as potential Doc downstream targets. The most severe thorax closure defect was observed by knocking down *arpc3B* and *βtub97EF*. *arpc3B* has been shown to play a role in organizing the actin cytoskeleton in yeast and *Drosophila* [2,61,62,63,64,65]. *βtub97EF* is essential for microtubule stability in the *Drosophila* gut [66]. We propose that the actin and microtubule cytoskeleton is required for thorax closure. Our knock-down analysis also identified three previously unidentified genes (*CG12164*, *CG14456*, and *CG16758*) as potential Doc target genes in thorax closure. *pcp* and *sdr* were upregulated in *Doc* hypomorphic wing discs and caused defective thorax development upon overexpression. While the presence of sequence motifs related to the Doc binding consensus in the upstream region of these genes is suggestive, it does not provide sufficient evidence for direct regulation.

### 3.4. The General Character between Dorasl Cloure, Thorax Clousre and Neural Tube Closure

Two processes are involved in the closure process: cell movement and fusion of two separated epithelium [3,39,40]. During the dorsal closure, the leading edge cells stretch along the dorsal-ventral axis by forming filopodial and lamellipodial protrusions [67,68], which are rich in F-actin [3]. The increase of F-actin pulls the whole epidermal forward over the amnioserosa surface. Under this force, the leading edge cells meet at the midline and begin to fuse together. The fusion process is guided by microtubule-dependent filopodia, which make sure the zippering of the opposing epithelial sheets and the correct matching of opposite segment [5,69,70]. The adherens junction and microtubule regulators are delivered to the leading-edge filopodia to make sure the adhesion of two adjoining epithelial sheets [71]. During the thorax closure, the movement of two epitheliums is achieved by the extension of F-actin which pulls the whole epithelium moving forward towards the midline. When two epitheliums are abutting, the adheres-type junctions are accumulated in leading edge cells, preparing for the adhesion of these two separated parts. The cytoskeleton also involved in the processes of thorax closure. Similarly, successful neural tube closure also involves fusion of the opposite neural folds to create the complete neural tube [72]. In all vertebrates, cellular protrusions, such as spike-like filopodia or sheet-like lamellipodia, are formed at the edge of the neural folds for the preparation of the neural fold attachments [73,74,75]. The components of adherens and tight junctions are also involved in the regulation of neural tube closure [76,77].

The potential role of Doc in regulating the cytoskeleton and microtubule dynamic or stability provides the general interests of epithelium fusion, including complete vertebrate neural tube formation, *Drosophila* embryo dorsal closure, and thorax closure. The collective cell movement and fusion of two separated epithelium [3,39,40] are the general character of forming a complete epithelium. Our study provides nice clue and candidate genes for epithelium formation in neural tube closure and wound healing. 

## 4. Materials and Methods

### 4.1. Drosophila Stocks

The following transgenes were used:

*ap-Gal4* (Bloomington #3041), 

*dpp-Gal4* (Milán and Cohen, 1999), 

*pnr-Gal4* (Bloomington #3039), 

*UAS-eyg* (Bloomington #26809), 

*UAS-pcp* (Bloominton #15367), 

*UAS-pnr* (Bloomington #7223), 

*UAS-sdr* (Kyoto Stock Center #206122), 

*UAS-tkv^DN^* (a gift from C. Dahmann), 

*UAS-arpc3B-RNAi* (VDRC #105278), 

*UAS-attc-RNAi* (VDRC #101213), 

*UAS-βTub56D-RNAi* (VDRC #109736), 

*UAS-βTub97EF-RNAi* (VDRC #105075), 

*UAS-CG12164-RNAi* (Tsinghua Fly Center #THU0252), 

*UAS-CG14456-RNAi* (VDRC #102147), 

*UAS-Doc1-RNAi* (VDRC #104927), 

*UAS-Doc2-RNAi* (VDRC #37634), 

*UAS-Doc3-RNAi* (VDRC #30550), 

*UAS-mec2-RNAi* (VDRC #104601), 

*UAS-pnr-RNAi* (VDRC #6224), 

*puc-lacZ* (P{lArB}puc[A251.1F3], Bloomington #11173).

Larvae were raised at 25 °C unless stated otherwise. Larvae of RNAi transgenes were raised at 29 °C for efficient expression. The genotype of all crosses is listed in the Appendix A.

### 4.2. Dissection of Larvae

Wing imaginal discs were dissected according to standard protocol and were fixed for 30 min in 4% paraformaldehyde in PBT (PBS with 0.3% Triton X-100).

### 4.3. Dissection of Pupae

The dissection protocol of pupae was modified from Martin-Blanco et al. [51]. The staged pupae were torn into an anterior and a posterior part. The latter, together with gut, salivary glands, central nervous system, and fat body was discarded. The anterior part with wing discs or notum still inside the pupal case was fixed for 1h in the 4% paraformaldehyde. Then, the ventral pupal case was removed from the fixed thorax followed by the immunohistochemistry steps. After the immunostaining, the fixed tissue was flattened with the prepupal wing and six leg disc orientated distally. Pupal staging is in hours after puparium formation (APF).

### 4.4. Immunohistochemistry

Fixed wing imaginal discs were stained with antibodies according to standard procedures. The primary antibodies used were: rabbit anti-Doc2, 1:2000 [30]; rabbit anti-pMad, 1:200 (Cell Signaling Technology #9516, Danvers, MA, USA); mouse anti-β-galactosidase, 1:2000 (Promega Z378B, Madison, WI, USA); Secondary antibodies used were goat anti-mouse DyLight 549, 1:200 (Agrisera AS09 643, Vännäs, Sweden) and goat anti-rabbit DyLight 488 1:200 (Agrisera AS09 637, Vännäs, Sweden). Images were collected using a Leica TCS-SP2-AOBS confocal microscope.

### 4.5. Adult Thorax Imaging

Adult thorax images were collected using microscope UV-CTS (Beijing, China).

### 4.6. Potential Target Genes Selection

The genes selected from the transcriptome analysis were used to blast with the Doc transcriptional factor binding site (TFBS, “TTCACACCT”, https://mccb.umassmed.edu/ffs/TFdetails.php?FlybaseID=FBgn0035956; accessed on 11 March 2022) in the 2 kb upstream region of the promotor. Twenty-seven of 177 genes were selected for the genetic screen based on the expression levels. The potential target genes were determined by the genetic screen crossing with the *pnr-Gal4* either with the RNAi lines or the UAS lines based on the expression trend. 

### 4.7. Thorax Closure Index

The distance between the two middle bristle was measured as the thorax closure index. For each *Drosophila*, three lines were drawn vertical to the thorax closure fusion mid-line for the measurement and at least five flies were used for quantification. 

## 5. Conclusions

We characterized the spatio-temporal changes of Dpp signaling in the leading edge cells during thorax closure detailing the role of Dpp in thorax closure. *Doc* was identified as a gene complex acting downstream of Dpp signaling by analyzing its expression pattern and its dependency on known Dpp pathway components. By molecular and genetic screens, nine genes were identified as potential Doc target genes with a role in thorax development (Figure 5).

## Figures and Tables

**Figure 1 ijms-23-04543-f001:**
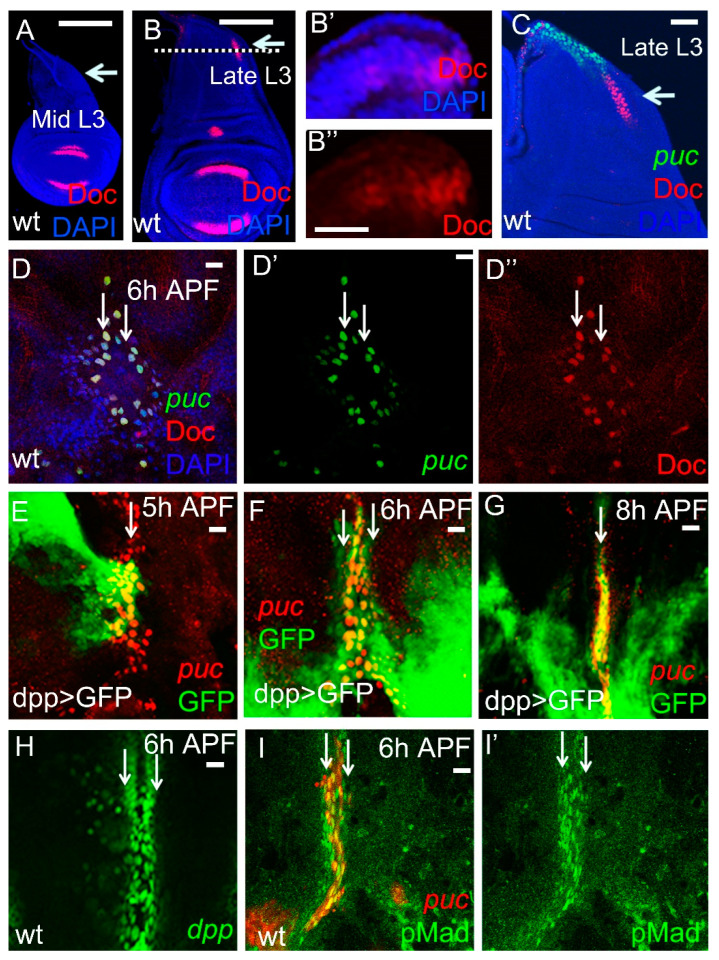
Dpp and Doc are expressed in the leading edge cells during *Drosophila* thorax closure. In this and subsequent figures, L3 wing discs and early pupal nota are shown. Other developmental stages are indicated. Wing discs are oriented with dorsal up and anterior left. Pupal nota are orientated with anterior up and the future fusion line in the middle. (**A**) At mid L3, *Doc* is not yet expressed in the posterior notum region. (**B**) At late L3, *Doc* expression arises in the future posterior notum. A cross view of the dotted line in (**B**) is shown in (**B’**,**B”**). *Doc* was only expressed in the columnar epithelium, but not in the peripodial epithelium cells. (**C**) At late L3, *Doc* and JNK are expressed in a complementary manner in the notum region, while at 6h APF (**D**), *puc-lacZ* (**D’**) and *Doc* (**D”**) are co-expressed in the leading edge cells. (**E**–**G**) *dpp* expression is revealed by *UAS-GFP* expression in the *dpp-Gal4* domain. Note that *GFP* is expressed in the leading edge cells (white arrows in **E**–**G**) throughout thorax closure from 5 to 8 h APF. *puc-lacZ* (red) is used as a marker of the leading edge cells. (**E**) At 5 h APF, the two separate heminota are moving toward the midline. Only a few puc-lacZ positive cells express *GFP*. (**F**) At 6 h APF, the two nota have established contact and start to fuse. (**G**) At 8 h APF, the two heminota have fused and form a continuous structure. GFP is still visible in the fusion midline. (**H**) *dpp-lacZ* expression (green) is observed in cells along the fusion midline. (**I**) pMad staining (green) (**I’**) indicates that Dpp signaling activity is high in the leading edge cells. The white arrows indicate the *Doc* expression region from 3rd instar larvae to 6 h APF. Scale bars, 2 μm.

**Figure 2 ijms-23-04543-f002:**
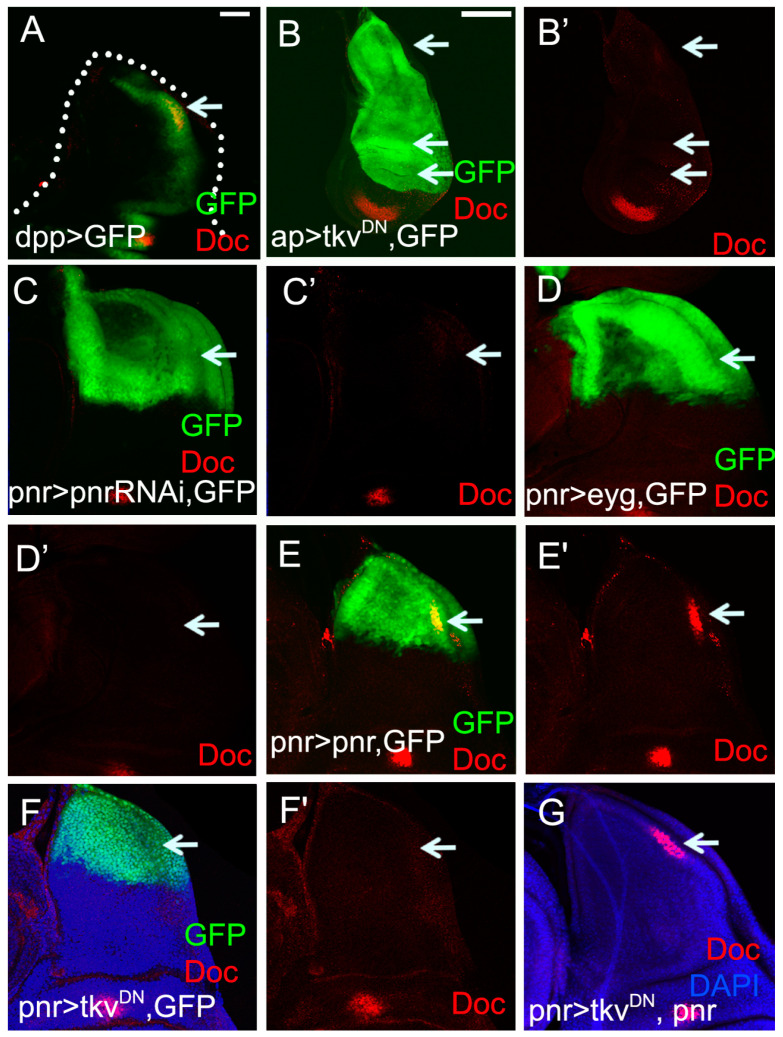
Doc is downstream of Dpp signaling. (**A**) *Doc* is expressed within the *dpp* expression region in the posterior notum of wing disc. The dotted line indicates the outline of the wing disc. (**B**) *Doc* is inhibited (**B’**) when a dominant negative form of Tkv (*tkv^DN^*) is expressed in the dorsal compartment by *ap-Gal4* driver (white arrows indicate the presumptive dorsal Doc domains). (**C**) *Doc* is inhibited (**C’**) when *pnr* is down-regulated. (**D**) *Doc* is suppressed (**D’**) when the *eyg* is expressed in the notal Pnr-domain. (**E** and **E’**) Overexpression of *pnr* does not affect notal *Doc* expression. (**F**) Notum *Doc* expression is reduced (**F’**) when *tkv^DN^* is expressed. (**G**) *Doc* repression is blocked when *pnr* is expressed in the *pnr > tkv^DN^* background. Scale bar, 2 μm.

**Figure 3 ijms-23-04543-f003:**
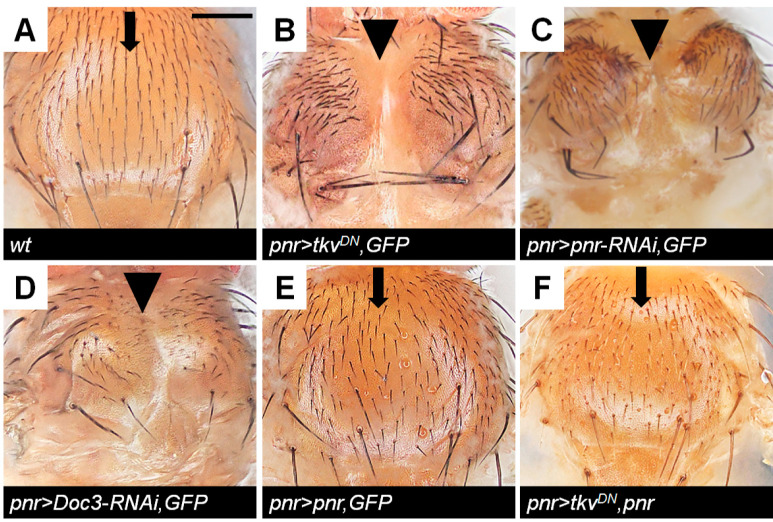
*Doc* is required for correct thorax closure. (**A**) Wild type adult notum. Arrow indicates the midline. (**B**) A medial cleft (arrowhead) develops at 100% frequency, indicating severe thorax closure defects, when Dpp signaling is inhibited by *tkv^DN^* over-expression. (**C**) Inhibition of *pnr* results in thorax closure defect with complete penetrance. (**D**) A mild thorax closure defect is also caused by *Doc3* knock-down. (**E**) Expression of *pnr* alone in the notum region does not induce thorax closure defects. (**F**) Expression of *pnr* rescues the thorax closure defect caused by *tkv^DN^*. The black arrows indicate the success of thorax closure. The black triangles indicate the failure of thorax closure. Scale bar, 200 μm.

**Figure 4 ijms-23-04543-f004:**
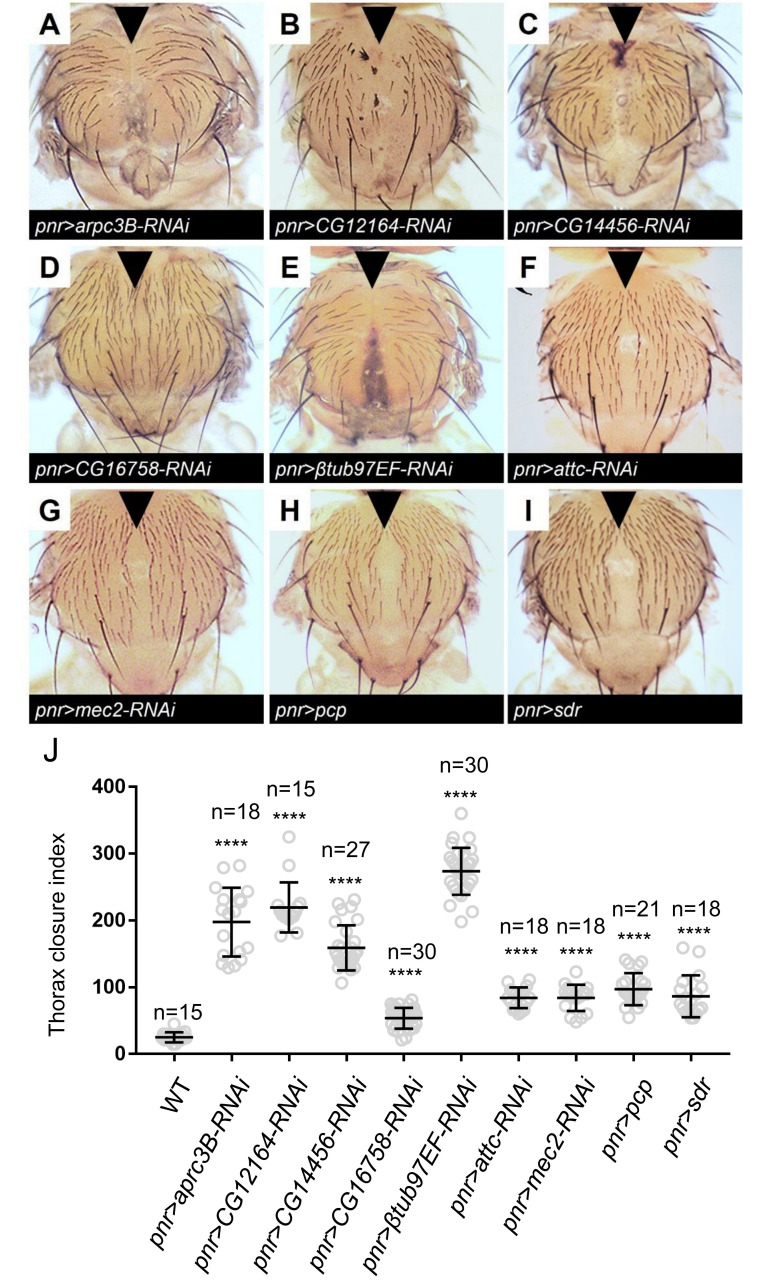
Adult thorax phenotype of nine potential Doc target genes. (**A**–**G**) Severe thorax closure defect when knocking down *arpc3B* (**A**), *CG12164* (**B**), *CG14456* (**C**), *CG16758* (**D**), *βtub97EF* (**E**), *attc* (**F**) and *mec2* (**G**). Severe thorax closure defect upon overexpression of *pcp* (**H**) and *sdr* (**I**). The black arrows indicate the success of thorax closure. The black triangles indicate the failure of thorax closure. Scale bar, 200 μm. (**J**) Quantification of the thorax closure severity indicated by the thorax closure index. Asterisks indicate the significant differences determined by Student’s *t* test. ****, *p* < 0.0001.

**Figure 5 ijms-23-04543-f005:**
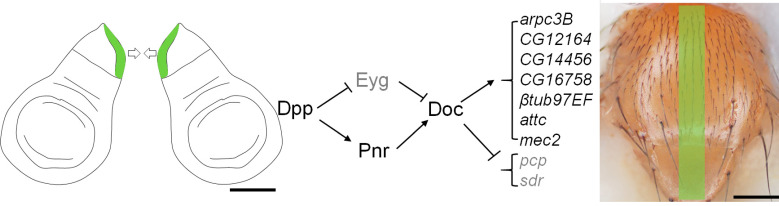
Summary of the signaling pathway regulates the thorax closure. Scale bars, 2 μm (**left**); 200 μm (**right**).

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
