# Peer review of "The Tbx6 Transcription Factor Dorsocross Mediates Dpp Signaling to Regulate Drosophila Thorax Closure"

_ijms, 2022, doi:10.3390/ijms23094543_

Round 1
Reviewer 1 Report
Brief summary
Thorax closure is a process that occurs during Drosophila metamorphosis and consists in the joining of the two wing imaginal discs to give rise to the adult thorax. It resembles vertebrate neural closure and Drosophila dorsal closure as it involves the closure of two separate epithelia. One of the main pathways involved in thorax closure is the Dpp pathway. Loss of Dpp signaling results both thorax and dorsal closure defects,
Dorsocross (Doc) transcription factors (Doc1, Doc2 and Doc3) are targets of Dpp in the embryo amnioserosa and in the embryonic ectoderm two tissues involved in embryonic dorsal closure. However, whether these TFs are involved either in dorsal or thorax closure has not been established. In this manuscript the authors explore the role of Doc in thorax closure. It has been shown that in the wing discs there are 4 doc expressing domains, two of which have been shown to be implicated in wing bending. Here, the authors investigate the role of one Doc expression domain in the posterior notum. They show that this domain is first detectable in late L3 wing discs and during early pupal development Doc colocalize with puc-LaZ (A251) a marker JNK activity, and Dpp in the leading edge cells. The authors then asked whether Doc is a target of the Dpp pathway in these cells. To do so they nicely blocked DPP signaling through a dominant negative form of thickvein and through Pnr RNAi and Eygone overexpression and consistently observed loss of Doc expression. Importantly, RNAi of Doc1 and 2 or Doc3 results in thorax closure defects. Finally, they selected 9 genes, presumably because they contained a Doc binding consensus motif ad showed that 7 were downregulated and two were upregulated upon Doc3 RNAi. They then show that RNAi of the 7 downregulated gene and overexpression of the two upregulated genes results in thorax closure defects.
Broad comments
In this manuscript the author nicely show through genetic epistasis experiments that Doc acts downstream Dpp to regulate thorax closure and implicate 9 Doc targets in this process. These findings could be extremely helpful to better understand closure processes and are definitively of interest to the scientific community working on this topic. However, it is not clear what is the relative contribution of each of the 3 Doc transcription factors (TF) in this process. The antibody that they use for the staining is an anti-Doc2 antibody, but they see thorax defects not only by depleting Doc2 (together with Doc1) but also by depleting Doc3. Additionally, the qPCR experiments on target genes were done only upon depletion of Doc3. The authors also do not explain how they identified the 9 Doc target genes. Are the three Doc factors redundant? The authors need to define this through the following experiment:
Specific comments
Major points:
- The authors talk about a Doc domain, but in the methods section they write that they used an anti-Doc2 specific antibody. Why then calling the domain Doc domain instead of Doc2 domain? Is the Doc expression domain in the posterior notum specific to Doc2? The authors should perform RNAi against each of the Doc TFs and verify if the staining is reduced.
- Are Doc mutants available to confirm the phenotype and could the phenotype be rescued by overexpressing either one of the Doc TFs?
- The authors show that Doc3 Rnai results in downregulation of 7 genes and upregulation of 2 genes upon Doc3 RNAi. Does RNAi of Doc1 and Doc2 also results in deregulation of these genes?
- The authors show that RNAi of the 7 downregulated Doc target genes and overexpression of the two upregulated genes results in thorax closure defects but they need to provide the quantification of the defect.
Minor points:
- The introduction is not very well organized. The author should provide more detail on the role of the dpp pathway in dorsal versus thorax closure and better explain the role of pannier and eyegone. It is difficult to distinguish between what is known about thorax closure and what is known about dorsal closure.
- The authors show that Doc3 Rnai results in downregulation of 7 genes and upregulation of 2 genes upon Doc3 RNAi. How did they select these targets? Is it on the basis of the presence of the Doc binding motif? If so they should state it clearly and they should add to the methods the way they identified the presence of the motif and how many genes contains this motif.
Author Response
Major points:
- The authors talk about a Doc domain, but in the methods section they write that they used an anti-Doc2 specific antibody. Why then calling the domain Doc domain instead of Doc2 domain? Is the Doc expression domain in the posterior notum specific to Doc2? The authors should perform RNAi against each of the Doc TFs and verify if the staining is reduced.
The three Doc genes have been demonstrated as functional redundant genes based on the similarities in sequence and expression as well as genetic analysis in previous studies [1,2]. Is these studies, the authors have demonstrated that the three genes have identical expression patterns expect the relative expression levels. In addition, the phenotype caused by different Doc-RNAi are the same only with different severity. Based on these data, the three Doc genes/proteins are collectively refer to ‘Doc genes/proteins’ in our study and the other groups. We have mentioned the redundancy function in the manuscript. We also add the statements to make this clear and to avoid confusion (highlight sentences).
- Are Doc mutants available to confirm the phenotype and could the phenotype be rescued by overexpressing either one of the Doc TFs?
The Doc homozygous mutants are embryonic lethal. It is technically difficult to test the role of Doc TFs in rescuing the Doc mutants in the thorax closure phenotype.
- The authors show that Doc3 Rnai results in downregulation of 7 genes and upregulation of 2 genes upon Doc3 RNAi. Does RNAi of Doc1 and Doc2 also results in deregulation of these genes?
The RNAi of Doc1 and Doc2 also deregulated these target genes, which strengthened the well-known redundant role of the three Doc genes on the other side.
- The authors show that RNAi of the 7 downregulated Doc target genes and overexpression of the two upregulated genes results in thorax closure defects but they need to provide the quantification of the defect.
All the RNAi of these target genes shows thorax closure defect. So we do not provide the quantification results to this closure defects.
Minor points:
- The introduction is not very well organized. The author should provide more detail on the role of the dpp pathway in dorsal versus thorax closure and better explain the role of pannier and eyegone. It is difficult to distinguish between what is known about thorax closure and what is known about dorsal closure.
Thank you for your suggestions. We reorganized the Introduction section of the manuscript to explain the role of pannier and eyegone more clearly. In addition, we reorganized the statement to distinguish the function of dorsal closure and thorax closure.
- The authors show that Doc3 Rnai results in downregulation of 7 genes and upregulation of 2 genes upon Doc3 RNAi. How did they select these targets? Is it on the basis of the presence of the Doc binding motif? If so they should state it clearly and they should add to the methods the way they identified the presence of the motif and how many genes contains this motif.
Thank you for your kind suggestion. The selected genes were screened by the Doc binding motif and tested by expression level in the Doc-RNAi background. Total 177 genes were screened based on the Doc binding motif and 27 genes were further screened based on the expression level changes in Doc-RNAi background. All of these 27 genes were used to test the possibility of thorax closure defect by genetic screen. Finally nine genes were defined as the potential target genes in regulating the thorax closure with the aberrant thorax phenotype. Following your suggestions, we add this information in the Methods section and the corresponding results section.
- Hashimoto, H.; Robin, F.B.; Sherrard, K.M.; Munro, E.M. Sequential Contraction and Exchange of Apical Junctions Drives Zippering and Neural Tube Closure in a Simple Chordate. Developmental Cell 2015, 32, 241-255.
- Liyuan; Sui; Gert; Pflugfelder; Jie; Shen. The Dorsocross T-box transcription factors promote tissue morphogenesis in the Drosophila wing imaginal disc. Development 2012.
Reviewer 2 Report
This article is well written, comprehensive and logically organized, and contains valuable information on the Tbx6 transcription factor Dorsocross mediates Decapentaplegic signaling to regulate Drosophila thorax closure. The authors did excellent research on the identification of T-box transcription factor genes Dorsocross (Doc) as Decapentaplegic (Dpp) targets in the leading edge cells of the notum in the late third instar larval and early pupal stages. The authors demonstrated that nine genes as potential Doc target genes with a role in thorax development y molecular and genetic screens. The submitted manuscript has significant scientific insights and the conclusions are soundly supported by the experimental data. Therefore, the present submission should be published in the future edition in the Special Issue of Feature Papers in Molecular Biology of the International Journal of Molecular Sciences.
Author Response
This article is well written, comprehensive and logically organized, and contains valuable information on the Tbx6 transcription factor Dorsocross mediates Decapentaplegic signaling to regulate Drosophila thorax closure. The authors did excellent research on the identification of T-box transcription factor genes Dorsocross (Doc) as Decapentaplegic (Dpp) targets in the leading edge cells of the notum in the late third instar larval and early pupal stages. The authors demonstrated that nine genes as potential Doc target genes with a role in thorax development y molecular and genetic screens. The submitted manuscript has significant scientific insights and the conclusions are soundly supported by the experimental data. Therefore, the present submission should be published in the future edition in the Special Issue of Feature Papers in Molecular Biology of the International Journal of Molecular Sciences.
Thank you for your positive comments and interests in our manuscript.
Reviewer 3 Report
Drosophila thorax closure in the pupal stage is one of the models used successfully in studies of the mechanism of tissue fusion processes. Decapentaplegic (Dpp) was documented previously as a transforming growth factor essential for thorax closure. However, downstream factors mediating the role of Dpp signaling in thorax closure were not identified. The Dorsocross (Doc) genes are expressed under the control of Dpp. The Doc gene complex encodes three functionally highly redundant transcription factors: Doc1, Doc2, and Doc3 belonging to the Tbx6 subfamily.
Authors investigated the role of Doc expression in the posterior notum of D. melanogaster wing discs, which gives rise to the adult thorax of the fruit fly and showed that the posterior notum expression domain corresponds to the future leading edge of the fusion discs. Doc expression required both Dpp and pannier (Pnr) activity. When Doc was reduced in the leading edge cells of the notum, a thorax closure defect was induced similarly to the phenotype caused by down-regulated Dpp signaling. Genetic analysis identified nine genes as downstream targets of Doc in regulating thorax closure.
The authors presented that the most severe thorax closure defect was observed by knocking down arpc3B and βtub97EF. The authors also proposed that the actin and microtubule cytoskeleton are required for thorax closure. In addition three previously unidentified genes (CG12164, CG14456, and CG16758) as potential target genes in thorax closure were identified.
The presented paper is good quality and I have only minor comments.
- Authors should be consequent in using italic for genes
- Figure 1 legend should be corrected
Author Response
Drosophila thorax closure in the pupal stage is one of the models used successfully in studies of the mechanism of tissue fusion processes. Decapentaplegic (Dpp) was documented previously as a transforming growth factor essential for thorax closure. However, downstream factors mediating the role of Dpp signaling in thorax closure were not identified. The Dorsocross (Doc) genes are expressed under the control of Dpp. The Doc gene complex encodes three functionally highly redundant transcription factors: Doc1, Doc2, and Doc3 belonging to the Tbx6 subfamily.
Authors investigated the role of Doc expression in the posterior notum of D. melanogaster wing discs, which gives rise to the adult thorax of the fruit fly and showed that the posterior notum expression domain corresponds to the future leading edge of the fusion discs. Doc expression required both Dpp and pannier (Pnr) activity. When Doc was reduced in the leading edge cells of the notum, a thorax closure defect was induced similarly to the phenotype caused by down-regulated Dpp signaling. Genetic analysis identified nine genes as downstream targets of Doc in regulating thorax closure.
The authors presented that the most severe thorax closure defect was observed by knocking down arpc3B and βtub97EF. The authors also proposed that the actin and microtubule cytoskeleton are required for thorax closure. In addition three previously unidentified genes (CG12164, CG14456, and CG16758) as potential target genes in thorax closure were identified.
The presented paper is good quality and I have only minor comments.
- Authors should be consequent in using italic for genes
Thank you for your suggestions. We have checked and corrected the font style of the genes in the manuscript.
- Figure 1 legend should be corrected
Thank you for your suggestions. We modify the legend of Figure 1 to make it more accurate.
Reviewer 4 Report
In their manuscript the authors investigated the molecular aspects of a very fascinating developmental biology issue: the closure of two separate epithelia, a process common also to tissue repair and crucial for morphogenesis. Such process is genetically tightly regulated by dozens of genes. In their scrutiny the authors first assess that in the Drosophila posterior notum the Doc expression is restricted to the future leading edge of the fusion discs. Afterwards, they observed that Doc expression is downstream of the TGFB signaling and of the GATA transcription factor pnr. Eventually, by mean of a genetic screen they identified nine putative Doc target genes, of which seven were down- and two up-regulated. Overall I appreciated the manuscript, though before publication there are a number of issues that have to be clarified.- the authors did not state on which strain the genetic screen has been carried out. I guess w.t. fly, am I wrong? Please provide details on this point.
- I guess that the library they used for the screen is from Vienna Drosophila Resource Center, am I worong? The authors should specify which library did they use to carry-out the screening, and how did they select the strains? Based on which readout? Were flies visually screened based on their aberrant thorax phenotype? The manuscript should be supplemented with all these details.
- The authors showed that all the nine genes identified upon the screening harbor a putative Doc consensus binding site within the 2 kb upstream region. Of what? ATG? transcription starting site? Please specify those details. Do all the nine genes harbor only a single Doc consensus binding site? or more than one? It seems just a single one, is it not? However, this point should be highlighted.
- Do those genes indicated by CG.... (e.g. CG12164, CG16758, etc...) encode for any protein displaying any similarity with mammalian proteins? In other words, do they have mammalian orthologs?
- Please at least validate a couple of down-regulated genes one of the up to provide a kind of proof of concept that they are "bona-fide" Doc target genes.
- Legend to figure 1 has to be detailed. E.g. what do bars indicate? and what dotted line (panel B)? etc..
Author Response
In their manuscript the authors investigated the molecular aspects of a very fascinating developmental biology issue: the closure of two separate epithelia, a process common also to tissue repair and crucial for morphogenesis. Such process is genetically tightly regulated by dozens of genes. In their scrutiny the authors first assess that in the Drosophila posterior notum the Doc expression is restricted to the future leading edge of the fusion discs. Afterwards, they observed that Doc expression is downstream of the TGFB signaling and of the GATA transcription factor pnr. Eventually, by mean of a genetic screen they identified nine putative Doc target genes, of which seven were down- and two up-regulated. Overall I appreciated the manuscript, though before publication there are a number of issues that have to be clarified.
Thank you for your positive comments and interests in our manuscript.
- The authors did not state on which strain the genetic screen has been carried out. I guess w.t. fly, am I wrong? Please provide details on this point.
The genetic screen was carried out on the pnr-Gal4 background. All selected RNAi lines were crossed with the pnr-Gal4 fly to validate the potential Doc target genes in regulating thorax closure. We add this statement in the section 2.4 of the manuscript.
- I guess that the library they used for the screen is from Vienna Drosophila Resource Center, am I wrong? The authors should specify which library did they use to carry-out the screening, and how did they select the strains? Based on which readout? Were flies visually screened based on their aberrant thorax phenotype? The manuscript should be supplemented with all these details.
The selected genes were screened by the Doc binding motif and tested by expression level in the Doc-RNAi background. Total 177 genes were screened based on the Doc binding motif and 27 genes were further screened based on the expression level changes in Doc-RNAi background. All of these 27 genes were used to test the possibility of thorax closure defect by genetic screen. The information of fly lines are listed in the Materials and Methods section. Finally the nine genes were defined as the potential target genes in regulating the thorax closure with the aberrant thorax phenotype. Following your suggestions, we add this information in the Methods section and the corresponding results section.
- The authors showed that all the nine genes identified upon the screening harbor a putative Doc consensus binding site within the 2 kb upstream region. Of what? ATG? transcription starting site? Please specify those details. Do all the nine genes harbor only a single Doc consensus binding site? or more than one? It seems just a single one, is it not? However, this point should be highlighted.
Thank you for your suggestions. The putative Doc consensus binding site was within the 2 kb upstream region of the transcription starting site. We also screened the genes with the sequence 2 kb upstream region of the ATG and got the same results. The Doc consensus binding site “TTCACACCT” was used to screen the potential target genes. We added this statement in the manuscript.
- Do those genes indicated by CG.... (e.g. CG12164, CG16758, etc...) encode for any protein displaying any similarity with mammalian proteins? In other words, do they have mammalian orthologs?
Based on the information on the flybase, CG12164 and CG14456 have no identified mammalian ortholog. CG16758 has the mammalian ortholog PNP. PNP deficient patients suffer from neurologic defects, such as cerebral palsy, ataxia and developmental delay.
- Please at least validate a couple of down-regulated genes one of the up to provide a kind of proof of concept that they are "bona-fide" Doc target genes.
The down-regulated genes and up-regulated genes are verified using qRT-PCR.
- Legend to figure 1 has to be detailed. E.g. what do bars indicate? and what dotted line (panel B)? etc.
Thank you for your suggestions. Bars indication has been added. The dotted line in panel B is the indication of the cross view of panel B’ which has been illustrated in the figures legend.
Round 2
Reviewer 1 Report
Rewiever Response: The authors improved the introduction and replied in a satisfactory way to some of my concerns. However, some points were not addressed, as discussed below.
Major points:
- The authors talk about a Doc domain, but in the methods section they write that they used an anti-Doc2 specific antibody. Why then calling the domain Doc domain instead of Doc2 domain? Is the Doc expression domain in the posterior notum specific to Doc2? The authors should perform RNAi against each of the Doc TFs and verify if the staining is reduced.
The three Doc genes have been demonstrated as functional redundant genes based on the similarities in sequence and expression as well as genetic analysis in previous studies [1,2]. Is these studies, the authors have demonstrated that the three genes have identical expression patterns expect the relative expression levels. In addition, the phenotype caused by different Doc-RNAi are the same only with different severity. Based on these data, the three Doc genes/proteins are collectively refer to ‘Doc genes/proteins’ in our study and the other groups. We have mentioned the redundancy function in the manuscript. We also add the statements to make this clear and to avoid confusion (highlight sentences).
Rewiever Response: Although in general, Doc proteins might have similar expression patterns and redundant roles, RNAi of doc3 alone does cause a closure defect, indicating that doc1 and doc2 cannot fully compensate for doc3 depletion. It is possible that this is due to a dosage effect, as also when both doc2 and 1 are depleted doc3 cannot compensate. So possibly when you go below a certain threshold of doc proteins then you have a phenotype. Is this the case? Since the authors have the tools, they need to verify that the doc domain signal is reduced upon doc 1 and 2 RNAi and doc3 RNAi. It is an important control. Furthermore in Line 112 , they should point out that they are using a Doc2 antibody.
Line 112: “Doc2 antibody staining was used as a proxy to identify Doc proteins expression pattern.”
- Are Doc mutants available to confirm the phenotype and could the phenotype be rescued by overexpressing either one of the Doc TFs?
The Doc homozygous mutants are embryonic lethal. It is technically difficult to test the role of Doc TFs in rescuing the Doc mutants in the thorax closure phenotype.
Rewiever Response: Ok, the lethality complicates the rescue experiment. I suppose that in early embryonic development doc proteins might not have redundant functions.
- The authors show that Doc3 Rnai results in downregulation of 7 genes and upregulation of 2 genes upon Doc3 RNAi. Does RNAi of Doc1 and Doc2 also results in deregulation of these genes?
The RNAi of Doc1 and Doc2 also deregulated these target genes, which strengthened the well-known redundant role of the three Doc genes on the other side.
Rewiever Response: I do not see these data in the manuscript. They need to be shown in the supplementary figures.
- The authors show that RNAi of the 7 downregulated Doc target genes and overexpression of the two upregulated genes results in thorax closure defects but they need to provide the quantification of the defect.
All the RNAi of these target genes shows thorax closure defect. So we do not provide the quantification results to this closure defects.
Rewiever Response: Ok, then indicate in the results or in the figure how many flies you looked at (N=x)
Minor points:
- The introduction is not very well organized. The author should provide more detail on the role of the dpp pathway in dorsal versus thorax closure and better explain the role of pannier and eyegone. It is difficult to distinguish between what is known about thorax closure and what is known about dorsal closure.
Thank you for your suggestions. We reorganized the Introduction section of the manuscript to explain the role of pannier and eyegone more clearly. In addition, we reorganized the statement to distinguish the function of dorsal closure and thorax closure.
Rewiever Response: The authors did a good job and now the introduction is clearer. I would just point out, if I understood correctly, that so far a role for pannier and eygone in thorax closure had not been reported. And conclude by saying:
“Therefore, downstream factors mediating the role of Dpp signaling in thorax closure have not yet been identified. »
Instead of
« However, downstream factors mediating the role of Dpp signaling in thorax closure have not yet been identified. »
- The authors show that Doc3 Rnai results in downregulation of 7 genes and upregulation of 2 genes upon Doc3 RNAi. How did they select these targets? Is it on the basis of the presence of the Doc binding motif? If so they should state it clearly and they should add to the methods the way they identified the presence of the motif and how many genes contains this motif.
Thank you for your kind suggestion. The selected genes were screened by the Doc binding motif and tested by expression level in the Doc-RNAi background. Total 177 genes were screened based on the Doc binding motif and 27 genes were further screened based on the expression level changes in Doc-RNAi background. All of these 27 genes were used to test the possibility of thorax closure defect by genetic screen. Finally nine genes were defined as the potential target genes in regulating the thorax closure with the aberrant thorax phenotype. Following your suggestions, we add this information in the Methods section and the corresponding results section.
Rewiever Response: This is an important piece of data and this information needs to be included in the results not just in the methods. Was the transcriptomic analysis published in another paper? If yes please cite the paper if not the experiments need to be described and the accession number of the transcriptomics analysis in NCBI need to be provided.
Author Response
- The authors talk about a Doc domain, but in the methods section they write that they used an anti-Doc2 specific antibody. Why then calling the domain Doc domain instead of Doc2 domain? Is the Doc expression domain in the posterior notum specific to Doc2? The authors should perform RNAi against each of the Doc TFs and verify if the staining is reduced.
The three Doc genes have been demonstrated as functional redundant genes based on the similarities in sequence and expression as well as genetic analysis in previous studies [1,2]. Is these studies, the authors have demonstrated that the three genes have identical expression patterns expect the relative expression levels. In addition, the phenotype caused by different Doc-RNAi are the same only with different severity. Based on these data, the three Doc genes/proteins are collectively refer to ‘Doc genes/proteins’ in our study and the other groups. We have mentioned the redundancy function in the manuscript. We also add the statements to make this clear and to avoid confusion (highlight sentences).
Rewiever Response: Although in general, Doc proteins might have similar expression patterns and redundant roles, RNAi of doc3 alone does cause a closure defect, indicating that doc1 and doc2 cannot fully compensate for doc3 depletion. It is possible that this is due to a dosage effect, as also when both doc2 and 1 are depleted doc3 cannot compensate. So possibly when you go below a certain threshold of doc proteins then you have a phenotype. Is this the case? Since the authors have the tools, they need to verify that the doc domain signal is reduced upon doc 1 and 2 RNAi and doc3 RNAi. It is an important control. Furthermore in Line 112 , they should point out that they are using a Doc2 antibody.
Line 112: “Doc2 antibody staining was used as a proxy to identify Doc proteins expression pattern.”
>Thank you for your suggestions. It is a pity that we could not test the expression level or potential threshold of Doc genes in the notum because only the Doc2 antibody is available in the original lab of Dr. Reim. We added the statement in our manuscript to discuss the potential threshold role of Doc in the notum and added the sentence as you suggested.
- Are Doc mutants available to confirm the phenotype and could the phenotype be rescued by overexpressing either one of the Doc TFs?
The Doc homozygous mutants are embryonic lethal. It is technically difficult to test the role of Doc TFs in rescuing the Doc mutants in the thorax closure phenotype.
Rewiever Response: Ok, the lethality complicates the rescue experiment. I suppose that in early embryonic development doc proteins might not have redundant functions.
>Previous studies have demonstrated the redundant roles of Doc in stage 13 of embryo. Right now, we have no idea of the redundant function in the early embryonic development stages.
- The authors show that Doc3 Rnai results in downregulation of 7 genes and upregulation of 2 genes upon Doc3 RNAi. Does RNAi of Doc1 and Doc2 also results in deregulation of these genes?
The RNAi of Doc1 and Doc2 also deregulated these target genes, which strengthened the well-known redundant role of the three Doc genes on the other side.
Rewiever Response: I do not see these data in the manuscript. They need to be shown in the supplementary figures.
>We added this part of data in the supplementary data, in which all of the up-regulated and down-regulated genes are shown.
- The authors show that RNAi of the 7 downregulated Doc target genes and overexpression of the two upregulated genes results in thorax closure defects but they need to provide the quantification of the defect.
All the RNAi of these target genes shows thorax closure defect. So we do not provide the quantification results to this closure defects.
Rewiever Response: Ok, then indicate in the results or in the figure how many flies you looked at (N=x)
>Thank you for your suggestions. For each genotype, over 50 adult flies were observed and up to 5 adult thorax were taken photos for record.
Minor points:
- The introduction is not very well organized. The author should provide more detail on the role of the dpp pathway in dorsal versus thorax closure and better explain the role of pannier and eyegone. It is difficult to distinguish between what is known about thorax closure and what is known about dorsal closure.
Thank you for your suggestions. We reorganized the Introduction section of the manuscript to explain the role of pannier and eyegone more clearly. In addition, we reorganized the statement to distinguish the function of dorsal closure and thorax closure.
Rewiever Response: The authors did a good job and now the introduction is clearer. I would just point out, if I understood correctly, that so far a role for pannier and eygone in thorax closure had not been reported. And conclude by saying:
“Therefore, downstream factors mediating the role of Dpp signaling in thorax closure have not yet been identified. »
Instead of
« However, downstream factors mediating the role of Dpp signaling in thorax closure have not yet been identified. »
>Thank you for your suggestions. We revised this sentence as you suggested.
- The authors show that Doc3 Rnai results in downregulation of 7 genes and upregulation of 2 genes upon Doc3 RNAi. How did they select these targets? Is it on the basis of the presence of the Doc binding motif? If so they should state it clearly and they should add to the methods the way they identified the presence of the motif and how many genes contains this motif.
Thank you for your kind suggestion. The selected genes were screened by the Doc binding motif and tested by expression level in the Doc-RNAi background. Total 177 genes were screened based on the Doc binding motif and 27 genes were further screened based on the expression level changes in Doc-RNAi background. All of these 27 genes were used to test the possibility of thorax closure defect by genetic screen. Finally nine genes were defined as the potential target genes in regulating the thorax closure with the aberrant thorax phenotype. Following your suggestions, we add this information in the Methods section and the corresponding results section.
Rewiever Response: This is an important piece of data and this information needs to be included in the results not just in the methods. Was the transcriptomic analysis published in another paper? If yes please cite the paper if not the experiments need to be described and the accession number of the transcriptomics analysis in NCBI need to be provided.
>Thank you for your suggestions. We added the transcriptomic analysis data in the supplementary table which includes all the regulated genes under Doc-RNAi treatment. We are sorry that the storage device for the raw data of the transcriptome sequence was missing during laboratory movement. It is a pity that we could not upload the raw data to NCBI. We provided the transcriptomic analysis report as alternative.
Reviewer 4 Report
I thank very much the authors for the efforts made in reviewing the manuscript and their exhaustive reply. I consider the revised version of the manuscript suitable for publication in IJMS.
Author Response
I thank very much the authors for the efforts made in reviewing the manuscript and their exhaustive reply. I consider the revised version of the manuscript suitable for publication in IJMS.
>Thank you for your positive comments.